# Subcellular Proteomics to Elucidate Soybean Response to Abiotic Stress

**DOI:** 10.3390/plants12152865

**Published:** 2023-08-04

**Authors:** Xin Wang, Setsuko Komatsu

**Affiliations:** 1College of Agronomy and Biotechnology, China Agricultural University, Beijing 100193, China; 020350286@163.com; 2Faculty of Environmental and Information Sciences, Fukui University of Technology, Fukui 910-8505, Japan

**Keywords:** soybean, abiotic stress, proteomics, subcellular

## Abstract

Climate change jeopardizes soybean production by declining seed yield and quality. In this review, the morphophysiological alterations of soybean in response to abiotic stress are summarized, followed by illustrations of cellular metabolisms and regulatory mechanisms to organellar stress based on subcellular proteomics. This highlights the communications associated with reactive oxygen species scavenging, molecular chaperones, and phytohormone signals among subcellular compartments. Given the complexity of climate change and the limitations of plants in coping with multiple abiotic stresses, a generic response to environmental constraints is proposed between calcium and abscisic acid signals in subcellular organelles. This review summarizes the findings of subcellular proteomics in stressed soybean and discusses the future prospects of subcellular proteomics for promoting the improvement of climate-tolerant crops.

## 1. Introduction

Soybean (*Glycine max*) is the most important leguminous crop with abundant vegetative oil and edible protein. Soybean oil containing an average of 181 g kg^−1^ alpha-linolenic acid in wild genotypes is beneficial to human health [1]. Physiologically active ingredients, especially isoflavones, are effective in promoting health and preventing disease [2]. With such great values, there is an increasing demand for soybeans and their byproducts. Therefore, improving soybean yield and quality is an important crop concern. However, around 30% of soybean production could decline by 2099 due to climate change, as estimated in [3]. Flood has reduced 39% and 77% of grain yield in the most flood-tolerant and -sensitive soybean cultivars [4], and drought has reduced 40% of grain yield [5]. An obvious yield loss of soybean was also observed under the presence of salt [6], extreme temperature [7], heavy metal [8], and other environmental stressors. Soybean yields are becoming alarming in light of climate change, and understanding plant response to abiotic stress is increasingly necessary.

Environmental stressors could be sensed by plant cells and then evoke status changes in subcellular components for cellular activities [9]. Among plant subcellular organelles, the nucleus has drawn more attention than others. Nuclear proteins, such as GmHsf34 [10], GmCaM4 [11], GmMYB84 [12], GmORG3 [13], GmERF135 [14], GmMYB14 [15], GmPKS4 [16], and GmNAC06 [17], enhance soybean tolerance to salt, drought, alkali, heat, and cadmium stress. Proteins located in or targeting the cell wall [18,19], plasma membrane [20,21,22], chloroplast [23,24], mitochondrion [25,26], and endoplasmic reticulum (ER) [27,28,29,30] play promising roles in soybean adaptation to abiotic stress. Subcellular organelles play a role in stress sensing and signaling, and they can alleviate metal toxicity through specific cellular compartments [18,24,30].

Omics approaches, such as transcriptomics and proteomics, could profile plant stress response based on changed genes and proteins, while only proteomics are available for high throughput analysis for organellar stress, because isolating RNA in specific organelle is challenging, and accuracy of subcellular prediction of identified transcripts is not as confident as expected. Therefore, subcellular proteomics with gene functional analysis will enrich our understanding of soybean organellar stress under environmental constraints. Subcellular proteomics in soybean and organellar stress under flood conditions have been summarized [31,32], while a systematical investigation into the organellar response to other environmental stressors is lacking. In addition, exploring a generic reaction of soybean in response to multiple environments is advantageous to breed stress resilience materials by manipulating the core cellular events.

In this review, we aim to elucidate soybean response to abiotic stress from the perspective of the subcellular level. To this end, the morphophysiological alternations in soybeans induced by climate change have been first summarized, and the main cellular events have been sketched. Then, the results of subcellular proteomics in soybeans under abiotic stress have been summarized to provide a deep illustration of organellar stress. On the basis of organellar stress-induced metabolisms, the contribution of proteins regulated in different subcellular compartments and associated with reactive oxygen species (ROS) homeostasis, protein folding, and phytohormone signals have been discussed, followed by a generic reaction initiated by organellar stress in stressed soybeans. This review not only characterizes the importance of subcellular proteomics to reveal organellar stress, but also outlines organellar stress signals to increase our ability to enhance soybean tolerance against climate change.

## 2. Morphophysiological Changes of Soybean under Abiotic Stress

Increased occurrence of extreme climate impedes agricultural production [33], and a series of morphophysiological alterations could prevent soybeans from stress injury [34]. In this section, soybean morphological alternations and physiological responses to adverse environments, such as flooding, drought, salt, and temperature stimuli, are sketched (Figure 1).

### 2.1. Morphophysiological Changes of Soybean under Flooding Stress

Global warming tends to induce extreme precipitation in many parts of the world [35]. Soybeans are vulnerable to excess water, and flooding stress influences seed germination [36] and seedling growth [37], causing a 16% yield loss globally [38]. The formations of adventitious roots, lateral roots, and aerenchyma, shoot elongation, petiole movement, stomatal conductance, and pigment accumulation are remarkably induced by hypoxia [39]. Based on these morphological changes, phenotypic variances of root system architecture [37], petiole length and angle [40], foliar damage [41], stem elongation rate [42], and plant survival rate [41] have been employed to screen flood-tolerant soybeans in greenhouses and fields. Physiological responses, such as RNA regulation, protein homeostasis, energy conservation, cell death, cell wall metabolism, and ROS homeostasis, are associated with soybean flood adaptation [39,43]. Interplays of ROS, energy depletion, photoinhibition, desiccation stress, phytohormone signal, C–N metabolism, and phenylpropanoid pathways are characterized during post-submergence conditions [44]. These results indicate that numerous morphophysiological alterations help soybeans adapt to flood and post-flood conditions.

### 2.2. Morphophysiological Changes of Soybean under Drought Stress

Approximately 40% of soybean seed loss is caused by drought [5]. Morphological alternations in root architecture [45], leaf senescence [46], and pod development [47] occur in drought-stressed soybean. The parameters of the root system, relative water content, leaf wilting, net photosynthesis efficiency, and biomass accumulation are utilized to explore drought-tolerant soybeans. Cell division [48], cell wall stability [48], phytohormone signals [48], oxidative defense [12,49], osmolyte and isoflavone biosynthesis [49,50], and signal transduction [50] are responsible for drought stress. Several transcription factors, including GmNFYA5 [51], GmNFYA13 [52], GmNFYb17 [53], GmVOZ1G [54], GmMYB84 [12], GmMYB14 [15], GmNAC085 [55], GmWRKY54 [56], and GsWRKY20 [57], improve drought tolerance by mediating stress-responsive genes independently or cooperatively.

### 2.3. Morphophysiological Changes of Soybean under Salt Stress

In soybeans, salt stress causes a decline in germination index and germination potential [58,59], growth retardation [60,61], leaf scorch [62,63], and yield loss [64,65], and these symptoms have been utilized for differing salt-tolerant variances. In addition, a heavier cotyledon is positively correlated to higher salt tolerance, indicating that cotyledon could serve as a non-destructive indicator for seedling salt tolerance [66]. Soybean salt tolerance has mainly been reviewed with ion homeostasis maintenance, osmotic response adjustments, and osmotic balance restoration [67]. Genes involved in ion accumulation [68,69,70,71,72,73], signal transduction [74,75,76], histone modification [77,78,79], antioxidant metabolism [80,81,82], phytohormone crosstalk [14,83], and flavonoid biosynthesis [84,85,86] could counteract the negative effects of salt stress. It showed that osmolyte accumulation, energy consumption, water retention, photoassimilate [87], abscisic acid (ABA) signal [52], and membrane permeability [88] were responsible for the salt–drought condition, while stress defense [16], antioxidant system [16], and DNA repairment [89] were activated under salt–alkali stress. Progressive achievements of soybean response to salt stress have been obtained, while since the episode of drought or alkali always happens in an occurrence with salt in fields, plant response to binary stimuli needs emphasizing.

### 2.4. Morphophysiological Changes of Soybean under Low-Temperature Stress

Low temperature, especially occurring at the emergency and end of the growing season of soybeans, limits their production. Cold stress at the flowering stage increases protein and ash contents, while decreasing fat concentration [90]. Cell membrane permeability was employed for seed germination as a physiological marker to assess soybean performance under low-temperature stress, and the number of active reaction centers and photosystem II effectiveness were used for potential yield [91]. Additionally, seed coat discoloration around the hilum region served to screen cold-tolerant soybeans in field assessment [92]. Our understanding of plant response to low-temperature stress is limited and mainly from the transcription factor CBF. *CBFs* are highly induced by low temperatures and could activate the accumulation of protective substances for cold acclimation in plants, such as osmolytes and cryoprotective proteins [93]. In soybeans, the inability of *CBF/DREB1* is consistent with cold intolerance, and downstream of *GmDREB1;1*-induced genes include ribosomal proteins and ABA receptors [94].

### 2.5. Morphophysiological Changes of Soybean under High-Temperature Stress

High temperature declines total dry matter, seed yield, and harvest index of soybeans [95]. Heat stress increased oleic acid accumulation, while decreasing linolenic/linoleic acid, raffinose, and stachyose contents in soybean seeds [96]. Higher levels of tocopherols, flavonoids, phenylpropanoids, and ascorbate precursors in seeds contributed to better germination for heat-tolerant soybeans compared to the sensitive genotype [97]. Tobacco plants overexpressing *GmGBP1* presented heat tolerance with better performance on seed germination, leaf growth, and plant survival through activating ABA or salicylic acid (SA) signals [98]. Overexpression of *GmHsp90A2* caused higher chlorophyll and lower malondialdehyde in soybean seedlings than the wild type by interaction with *GmHsp90A1* under heat stress [99]. For 3-day-old seedlings, even 3 h heat was sufficient to trigger significant changes at the protein level in root hairs and stripped roots, and stress-related proteins played roles in chromatin remodeling and post-transcription regulation in root hairs within 24 h exposure to heat [100]. Heat influences on reproductive development are more prominent than in the vegetative stage [101]. High temperature induced nighttime respiration rate, while decreasing photosynthesis and pod set percent compared with the optimum condition [102,103]. Heat waves in the early pod developmental stage imposed more impact on the reproductive process than photosynthesis efficiency, and soybeans could not rescue yield loss when stress occurred in the late reproductive stage [104]. The energy metabolism, protein metabolism, nitrogen assimilation, and ROS detoxification were activated in the anthers of heat-tolerant soybeans [105], and pollen germination and canopy reflectance at the visible spectrum could be used as a high throughput phenotypic tool to evaluate heat-tolerant cultivars [102].

### 2.6. Morphophysiological Changes of Soybean under Other Stresses

In addition to the stressors mentioned above, deficient nutrients and excess metal ions also disrupt soybean growth. For example, phosphate starvation limited soybean growth due to its low availability in soil [106]. Phytohormone, calcium, and MAPK signals conferred phosphate deficiency and salt stress through mediating gma-miR1691-3p, gma-miR5036, gma-miR862a, and gma-398a/b [107]. The application of silicon (0.5 mM) could alleviate plant retardation by zinc deficiency by facilitating zinc in root apoplast and transport to shoots [108]. Soybean response to metal stress, such as arsenic, aluminum, and cadmium, has also been investigated. Arsenic stress declined root absorption rate, reduced leaf xylem vessels, and induced abnormal stomata [8], while the application of nitric oxide and hydrogen peroxide could mitigate arsenic toxicity via ion sequestration to vacuoles and activated the ascorbate-glutathione cycle [109]. It showed that a low concentration (100 µM) of aluminum caused a relatively low accumulation of hydrogen peroxide/malondialdehyde and high soluble protein in root tips, and activated catalase was necessary for plant tolerance to high concentrations (200 and 400 µM) of aluminum [110]. In addition to catalase, GmWRKY81 positively regulated soybean tolerance to aluminum toxicity by activating aluminum transport and organic acid secretion [111]. Greater plant height, root length, root number, and root area were observed in drought-tolerant cultivars despite the higher cadmium in soybean roots, and the biosynthesis of auxin, gibberellic acid (GA), methyl jasmonate (MeJA), SA, and hydrogen peroxide, as well as cadmium transport, partly contributed to cadmium tolerance under drought [112]. These findings indicate that biosynthesis/signal of phytohormones, transport/sequestration of metal ions, and antioxidant metabolism could act synergistically in soybean adaptation to nutrient deprivation and metal stress.

## 3. Subcellular Response in Soybean under Abiotic Stress

Environmental stressors could be sensed in the cell surface or membrane and then alter cellular components for stress adaptation [9]. Subcellular response in the cell wall, plasma membrane, nucleus, ER, chloroplast, mitochondrion, Golgi, and peroxisome could be integrated to relay stress signals [113]. In this section, the subcellular proteomics of soybean under abiotic stress is summarized (Table 1). In addition, based on the results of subcellular proteomics and genetic studies, cellular events in response to organellar stress in stressed soybeans under environmental stimuli are sketched (Figure 2).

### 3.1. The Response of Endoplasmic Reticulum in Soybean under Abiotic Stress

Adverse environments disrupt protein folding in plants, resulting in the excessive accumulation of unfolded or misfolded proteins, referred to ER stress [123]. ER proteomics clarified that protein synthesis and glycosylation were suppressed in the root tips of flood-stressed soybeans compared to the untreated plants [114]. ER stress occurred with the decrease of protein disulfide isomerase-like proteins and heat shock proteins by flood conditions and drought, respectively, leading to dysfunction of protein folding and reduced accumulation of glycoproteins [27]. Flood and drought activated calcium release from ER, disturbing calcium homeostasis required for protein folding in ER [27]. Some ER proteins were implied in conferring soybean tolerance to abiotic stress. The ER-localized *GmSALT3* could improve soybean salt tolerance of near-isogenic lines by preventing ROS overload, calcium signal activating, vesicle trafficking, and diffusion barrier formation [75]. GmHMA3, mainly expressed in root ER, facilitated cadmium translocation into root ER and limited ion translocation from roots to shoots, indicating a high cadmium sensitivity of ER [28]. These studies suggest that ER could help soybeans adapt to abiotic stress through modulating protein metabolism, calcium signal, ROS scavenging, and metal ion transport.

### 3.2. The Response of Chloroplast in Soybean under Abiotic Stress

The chloroplast is the organelle where photosynthesis takes place and is a major site for ROS production. Abiotic stress declined net photosynthetic rate in soybean due to degradation of ribulose-1,5-bisphosphate carboxylase/oxygenase (RuBisCO) [124], reduction of chloroplast, grana number, leaf area [125], and increase of intercellular space [125]. Less formation of chloroplast protrusion and RuBisCO-containing body in mesophyll cell chloroplast of soybean leaves contributed to salt tolerance [124]. In salt-stressed roots, tetratricopeptide repeat domain-containing protein PYG7, CHAPERONE-LIKE PROTEIN OF POR1, outer envelope pore protein 24B, and TIC110, which were involved in chloroplast development, decreased; in leaves, plastid transcriptionally active 16, PRA1 family protein B5, chlorophyll *a/b* binding proteins 1 and 1.2, as well as RAN GTPase-activating protein 1, were downregulated, due to the winding down of photosynthetic activities in exchange for ramping up salt response [118]. Shading stress increased the number of chloroplasts in soybean leaves, while the size of chloroplasts and starch grains decreased compared with control, and decreased transmitting capacity from photosystem II to photosystem I was attributed to decreased photosynthetic capacity [126]. Accumulation of ascorbate peroxidase, catalase, and superoxide dismutase increased in the chloroplasts of ozone-stressed soybean leaves, and antioxidant defense was activated to cope with ROS overload [23]. These reports suggest that low-efficient photosynthesis is notable for chloroplast stress, and activated ROS scavenging could restore chloroplast homeostasis in soybean under stressful environments.

### 3.3. The Response of Mitochondria in Soybean under Abiotic Stress

Abiotic stress causes considerable impairments to soybean mitochondrion, leading to mitochondria stress. Flood conditions directly impaired electron transport chains in roots and hypocotyls of soybean seedlings, although mitochondrial NADH production increased through the tricarboxylic acid cycle [115]. In flooded roots with hypocotyl of soybean seedlings, the mitochondrion was targeted by aluminum oxide nanoparticles, which could minimize stress effects by maintaining membrane permeability and tricarboxylic acid cycle activity [25]. The performance of early germination and seedling establishment of soybeans with contrasting salt tolerance was examined with the finding of a general increase in mitochondrial respiration for the tolerant genotypes, indicating that the alternate oxidase pathway of mitochondrial respiration efficiently operated for salt improvement [127]. Drought inhibited root growth during seedling establishment, while seed priming with uniconazole significantly alleviated inhibition, increased the number of roots, and mitigated drought-induced damage to mitochondria through activation of the osmotic adjustment system and antioxidant activities [128]. Cadmium acted as an uncoupler of mitochondrial oxidative phosphorylation in soybean roots, and it disturbed cellular respiration and induced oxidative cellular stress [26]; however, the application of mitochondria-targeted antioxidant, MitoTEMPO, diminished cadmium-dependent induction of superoxide anion and lipid peroxidation, suggesting that mitochondria-derived ROS was engaged in metal uptake [129]. These studies indicate that ROS overload induced by abiotic stress is responsible for mitochondria stress.

### 3.4. The Response of Nucleus in Soybean under Abiotic Stress

The nucleus is the subcellular compartment containing nearly all of the genetic information, while abiotic stress disturbs the ultrastructure of the nucleus, leading to dysfunction of transcriptional regulation governed by nuclear. In soybean, compared with other abiotic stressful conditions, nuclear proteomics was widely used to explore flooding response. The initial flood reduced the accumulation of proteins related to the exon junction complex, pre-ribosomal biogenesis, and histone variants in nuclei of the soybean root tip, leading to growth retardation [116]. Nuclear proteins of zinc finger/BTB domain-containing protein 47, glycine-rich protein, and Rrp5, involved in the ABA signal, were phosphorylated by flood conditions, indicating that ABA is coordinately regulated with nuclei stress via protein phosphorylation [117]. The manganese toxicity significantly increased the abundance of GTP-binding nuclear protein Ran-3 and Ran-binding protein 1, which acted together in protein delivery across the nuclear membrane for soybean root growth [130]. These results indicate that chromatin structure, phytohormone response, and protein transport are in response to nuclei stress during soybean growth under environmental stressors.

### 3.5. The Response of Plasma Membrane in Soybean under Abiotic Stress

The plasma membrane is a dynamic compartment harboring almost all receptors in signal transduction [131]. Flood-induced plasma membrane stress in soybean roots and hypocotyls and flood-altered plasma membrane proteins were mainly categorized into protein destination/storage, energy catabolism, and stress defense [120]. Superoxidase dismutase, heat shock cognate protein, and signal-related proteins prevented plant cells from oxidative damage and ion imbalance in flooded soybeans [120]. Under aluminum stress, differentially accumulated proteins in the plasma membrane functioned in membrane trafficking, cell wall modification, defense response, and signal transduction [119]. Genes, which protect cells from plasma membrane stress, have also been reported. *GmAKT1*, which is an AKT-type K^+^ channel, positively maintained Na^+^/K^+^ homeostasis and upregulated the expression of genes involved in ion uptake under salt stress in soybeans [132], and *GmSOS1* enhanced soybean salt tolerance through alleviating ion imbalance [66]. The application of calcium could ameliorate the growth retardation of soybean under acid rain conditions and promote plant recovery from stress by enhancing the activity of plasma membrane H^+^-ATPase and nutrient uptake [133]. These findings show that membrane trafficking, ion homeostasis, and signal transduction are major cellular events of plasma membrane stress in soybeans under abiotic stress.

### 3.6. The Response of Cell Wall in Soybean under Abiotic Stress

The cell wall is the first subcellular organelle to sense and transmit stress signals into the cell interior, and thereby modify cellular metabolisms for plant adaptation to diverse environmental conditions. In soybean seedlings, along with suppressed lignification in flooded roots, the cell wall proteins, including two lipoxygenases, four germin-like proteins, three stem 28/31 kDa glycoprotein precursors, and one Cu–Zn superoxide dismutase, decreased under flood conditions, indicating that suppression of cell wall lignification attributed to the downregulation of ROS scavenging and JA biosynthesis [120]. High accumulation of pectin in the cell wall was necessary to improve soybean salt tolerance [134], and stress response, ion homeostasis, and cell wall modification, which were coordinately modified by the dramatic alteration of histone marks by salt priming, were noted [79]. A high expression of *GmEXPB2* in the cell wall of roots improved soybean tolerance to phosphorus starvation by promoting hairy root elongation [135]. Overexpression of *GmEXPB2*, whose expression was impacted by GmPTF1, was associated with better performance of root growth, plant biomass accumulation, and phosphorus uptake under phosphorus deficiency [136]. These studies show that high pectin and enhanced lignification are responsible for activated ROS scavenging, phytohormone signal, and adaptative root architecture for cell wall stress under adverse conditions.

### 3.7. The Response of Other Organelles in Soybean under Abiotic Stress

Plant peroxisome is a complex factory of antioxidant systems, and regulation of ROS in peroxisome could take place by post-translational modifications of stress-responsive proteins involved in ROS production or scavenging [137]. Cellular events of fatty acid beta-oxidation, glyoxylate cycle, photorespiratory glycolate metabolism, stress response, and metabolite transport were enriched by changed proteins in the peroxisomes of etiolated soybean cotyledons [121]. The accumulation of GmPNC1, the peroxisomal membrane protein, increased after germination in the darkness and decreased by illumination [122]. GmPUB6, a U-box gene in the peroxisome, functioned as a negative regulator for drought and osmotic tolerance, and the gain-of-function of GmPUB6 decreased plant survival, suppressed seed germination, and inhibited ABA-mediated stomatal conductance in Arabidopsis [138]. Vacuoles were separated into two different compartments of storage and lytic [139], and the grouping of vacuolar process enzymes into seed and vegetative types is related to vacuole classification [140]. The typical storage compartment is the protein storage vacuole found in seeds, while heat stress disturbs its structure and membrane integrity in cotyledonary parenchyma cells in soybeans [141]. Under drought stress, three cysteine proteases in vacuolar process enzymes were induced, and plants with decreased C1 cysteine protease activity had higher biomass and protein accumulation compared with wild-type soybeans [142].

## 4. Proteins Regulated Organellar Stress among Subcellular Compartments in Soybean under Abiotic Stress

Abiotic stress causes perturbation in subcellular organelles and induces organellar stress, which could provoke signal networks to restore cellular homeostasis [9]. With the development of subcellular proteomics, subcellular proteins, post-translational modifications of stress-responsive proteins, and protein–protein interactions in different subcellular compartments in soybean under flood conditions were previously reviewed [31,32]. In this section, subcellular proteins mainly related to ROS catabolism, protein folding, calcium signal, and ion homeostasis in stressed soybeans under harsh environments are updated (Figure 3).

### 4.1. Reactive Oxygen Species

Environmental stressors challenge cellular redox balance and lead to an increase in ROS production. ROS could be efficiently controlled by intracellular antioxidases and antioxidants, and subcellular proteins affected by oxidative stress in soybeans were investigated in the cell wall [120], plasma membrane [143], mitochondria [25,144,145], chloroplast [23], and ER [75]. In flooded soybeans, reduced accumulation of superoxide dismutase and germin-like proteins inhibited cell wall formation and lignin biosynthesis [120]. Plasma membrane contributed to cell wall construction, and flood-induced superoxide dismutase could protect soybean root (including hypocotyl) cells from oxidative damage [143]. In the mitochondria of flooded soybean roots, glutathione-*S*-transferase and lipoxygenase increased, along with an increase in hydrogen peroxidase accumulation [145], and the reduced activity of the ascorbate/glutathione cycle was related to the reduced activity of ROS scavenging upon exposure flooded soybeans to aluminum oxide nanoparticles [25]. Oxidative burst was induced by ozone stress, along with an increased accumulation of proteins involved in antioxidant defense and carbon metabolism, suggesting that the availability of sucrose was involved in antioxidative regulation [23]. Protein maturation in ER represents another source of ROS. GmSALT3, which is the ER-localized protein, prevented ROS overaccumulation in the roots of salt-stressed soybeans [75]. Compared with other subcellular organelles, a number of nuclear proteins, which were grouped in NAC [17,55], MYB [12,61,86,146], WRKY [57,111], and HSF [84] families, have been characterized to regulate hydrogen peroxidase production under osmotic stress induced by drought and salt.

### 4.2. Molecular Chaperons

Except for ROS burst, environmental stressors disturb protein homeostasis, leading to ER stress [123]. Molecular chaperones, along with calnexin and calreticulin, promote protein folding and are critically important in preventing misfolding and irreversible aggregation of proteins [147]. The soybean genome contains 61 *HSP70* genes, and their expression was inducible during plant growth and under heat and drought [148]. The calnexin, 70-kDa heat shock cognate protein, and luminal binding protein, which were either localized in ER or cytoplasm, decreased by flood conditions [114], and calcium release from ER risked the function of molecular chaperones [27]. Under flooding stress, there was a reduction of cell death in the mutant line compared with the wild-type soybean, and the comparative proteomic analysis found that calreticulin only accumulated in the mutant, indicating that regulation of cell death partly by glycoprotein folding was an important factor for flood-tolerance acquisition of soybean [29]. Not only in the ER, but heat shock cognate protein in the plasma membrane also increased, which could protect proteins from denaturation and degradation in flood-stressed soybeans [143]. GmHsp90A2 in the cytoplasm and cell membrane was a positive regulator to enhance soybean heat tolerance through increasing chlorophyll accumulation and decreasing malondialdehyde content [99]. These findings indicate that abundant molecular chaperones are necessary to protect cells from protein denaturation and ROS overload under abiotic stress.

### 4.3. Other Proteins

Subcellular proteins associated with photosynthesis efficiency [149], ion homeostasis [17,70,72,73], calcium signal [20,75], and metal export [30] could regulate soybean response to abiotic stress. GmPLAs were predicated either in the chloroplast, vacuole, cytoplasm, or extracellular, and loss-of-function of the two paralog genes *GmpPLA-IIε* and *GmpPLA-IIζ* altered root architecture under phosphorus deficiency and improved soybean tolerance to flood, drought, and iron deficiency [150]. Overexpression of *GmPIP2;9*, which is a plasma membrane protein, improved water transport activity in salt-stressed soybeans and consequently increased photosynthesis rate, stomata conductance, transpiration rate, pod number, and seed size compared with the wild type [149]. The plasma membrane protein GmSOS1 [73], Golgi-located protein GmNHX5 [72], and nuclear protein GmNAC06 [17] positively regulated salt tolerance by maintaining the balance of Na+/K+ ratio and the increase of proline/glycine betaine in soybeans. The ER-localized protein GmSALT3 improved soybean growth under the presence of salt by activating calcium signal, vesicle trafficking, and diffusion barrier formation [75]. Except for the salt-induced subcellular proteins mentioned above, proteins in response to metal stress have also been characterized. For example, 14 *GmMTPs* genes in soybean were identified, and *GmMTP8.1* in ER conferred manganese tolerance by stimulating manganese export into root vacuole [30].

## 5. Phytohormone Signals in Soybean under Abiotic Stress

Phytohormone signals, such as ABA, brassinosteroid, ethylene, GA, and SA, as well as their crosstalk, are inducible to modulate subcellular stress in soybeans (Figure 4). ABA could induce the formation of secondary aerenchyma in soybean hypocotyl under flooding stress [151], and the application of ABA improved soybean flooding tolerance through the control of energy conservation and the enhancement of cell wall lignification [152,153]. Nuclear proteins GmNFYA13 [52] and GmFBX176 [49] played a positive and negative role, respectively, in soybean tolerance to drought and salt by mediating plants’ sensitivity to ABA. *GmNFYA3*, which is the target of miR169, positively modulated soybean drought tolerance via the enhancement of ABA biosynthesis [154]. GmMYB14 could improve drought tolerance and high-density yield in soybean by activating the brassinosteroid signal [15]. Overexpression of *GmGBP1* in tobacco plants made a promotion of flowering, plant height, and heat tolerance, and the transcript level of *GmGBP1* was inducible by GA and SA [98]. The crosstalk among phytohormones showed that ABA and ethylene were activated to coordinate energy-saving processes under both flood conditions and drought; however, stimulated auxin response was flood specific, indicating that the auxin signal distinguished flood-specific regulation of stress responses [155]. These reports illuminate that phytohormone signals could improve soybean tolerance to abiotic stress via the interaction of energy catabolism, cell wall modification, and plant architecture.

## 6. Concluding Remarks

Demand for soybeans is increasing due to abundant protein and oil, while a dramatic yield loss is caused by climate change. Around 39–77% of grain yield was threatened by waterlogging [5]. Water deficit during the flowering and pod-setting stage caused a significant yield loss of 73–82% per plant, and the greatest loss in hundred-seed weight declined by around 42–48% by drought in the pod-filling stage [156]. Soybean yield sharply declined with increasing salt treatments, representing a 40% reduction when irrigation water salinity increased to EC around 7 dS m^−1^ [157]. For temperature stressors, nearly 24% and 27% yield loss was caused by cold [91] and heat [158], respectively. Therefore, soybeans have to endure abiotic stress attributed to yield loss during plant growth and development. Although morphophysiological alterations and organellar stress are essential for soybean adaptation to abiotic stress, our understanding of multiple environmental stressors, which always happen simultaneously in the field for crop production, is limited. A stress response is environment specific, while declined stomatal conductance, photosynthesis capacity, energy provision, activated ABA and calcium signals, and increased membrane permeability are common features of various environmental constraints. Calcium signal interacts with kinase, lipid-binding proteins, and ROS metabolic enzymes, eventually for ABA accumulation and stomatal closure in plants [9,159]. Based on the universal character of spiked calcium signal by abiotic stress and the current findings from soybean subcellular proteomics, it is reasonable to propose that calcium signal generated from the reservoirs inside plant cells could initiate the stress responses in an ABA-dependent manner, whereas calcium channels, especially in the chloroplast, peroxisome, mitochondrion, and ER, need fruitful investigation to elaborate the encoding mechanism of calcium signal (Figure 5).

Numerous proteins have been identified in enriched organellar, while large datasets derived from subcellular proteomics remain stored, because the function of most proteins is still unclear due to genetic redundancy in the soybean genome and experimental limits in functional validation. Bioinformatics are indispensable for data mining and interpretation for functional omics, and provide a base for collecting information for hub genes involved in plant stress responses [160]. In most cases, for omics data analysis, a cut-off of fold change greater than 1.5 is always used to screen the significance, while the rest are considered unchanged and put aside, which jeopardizes the identification of essential proteins with low abundance change in response to abiotic stress. System biology analyses, such as protein–protein interaction, enable a deeper understanding of regulatory networks and provide access to the molecular and mechanistic aspects of protein science, and interaction networks constructed on protein abundance are especially needed. Several bioinformatic databases available for crop science have been summarized [161], while proteomic databases, as well as their integration with other functional omics, are limited. Integrated omics are highlighted to reduce the false positives generated from the single data source and improve the statistical and biological area, and panomics, which have been proposed recently, could provide a platform to integrate complex omics. The integration of panomics with environmental platforms accelerates the identification of hub genes involved in stress response via deep learning, which is advantageous to molecular-assisted breeding and gene function analysis by genome editing [162]. Therefore, subcellular proteomics integrated with other functional omics can be available to generate a large-scale map to illustrate how molecular networks connect proteins and potential subcellular metabolisms to breed elite lineages with enhanced tolerance to abiotic stress.

## Figures and Tables

**Figure 1 plants-12-02865-f001:**
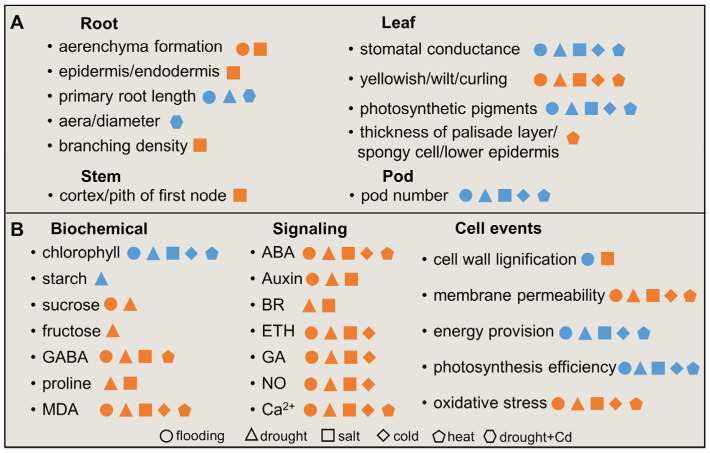
Overview of morphophysiological changes of soybeans under abiotic stress. (**A**) Morphological changes of soybeans under abiotic stress. (**B**) Physiological response of biochemical, signaling, and cellular events in soybean seedlings under abiotic stress. The circle, triangle, square, diamond, pentagon, and hexagon indicate flooding, drought, salt, cold, heat, and drought with cadmium, respectively. The orange and blue colors indicate an increase (activated) and decrease (suppressed) of examined parameters (metabolisms), respectively. ABA, abscisic acid; BR, brassinosteroid; ETH, ethylene; GA, gibberellic acid; GABA, gamma-aminobutyric acid; MDA, malondialdehyde.

**Figure 2 plants-12-02865-f002:**
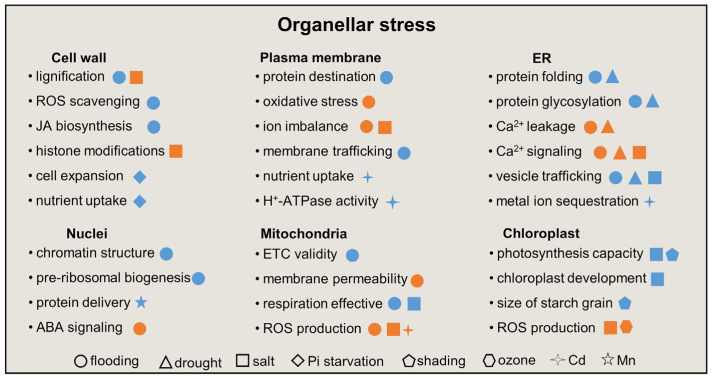
Overview of cellular metabolisms in response to organellar stress in soybeans under abiotic stress. The circle, triangle, square, diamond, pentagon, hexagon, cross star, and star indicate flooding, drought, salt, phosphorus starvation, shading, ozone, cadmium, and manganese, respectively. The orange and blue colors indicate an activation and suppression, respectively. ABA, abscisic acid; ER, endoplasmic reticulum; ETC, electron transport chain; JA, jasmonic acid; ROS, reactive oxygen species.

**Figure 3 plants-12-02865-f003:**
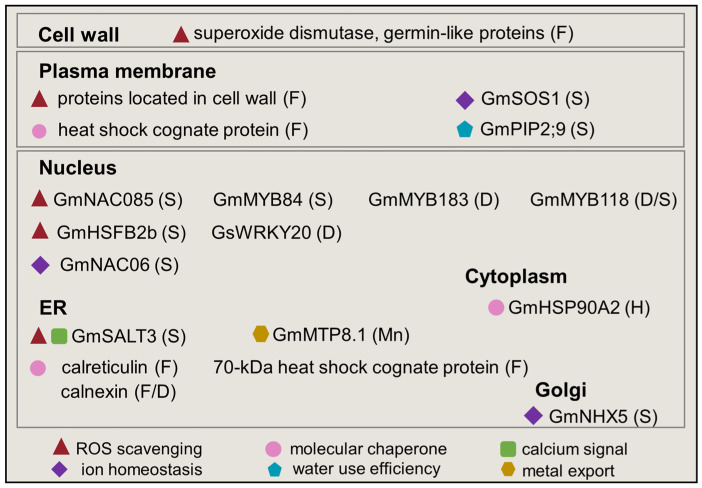
Overview of organellar proteins in soybeans under abiotic stress. Proteins recognized in ROS metabolism, molecular chaperone, calcium signal, ion homeostasis, water use efficiency, and metal export among different organelles in soybean plant cells under abiotic stress are indicated by a red triangle, pink circle, green square, purple diamond, blue pentagon, and brown hexagon, respectively. The letters in brackets indicate abiotic stress in which the organellar proteins are induced. F, D, S, and H mean flooding, drought, salt, and high temperature, respectively.

**Figure 4 plants-12-02865-f004:**
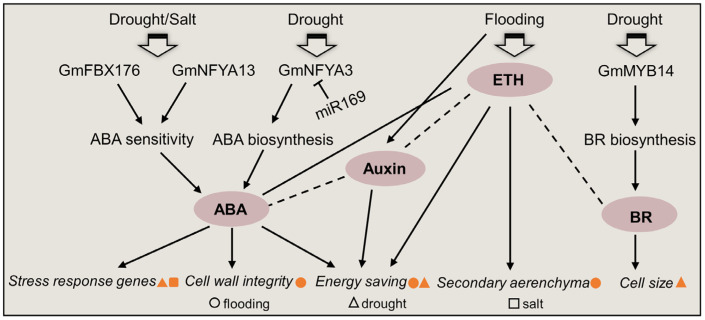
Overview of phytohormone signals in soybeans under abiotic stress. ABA, abscisic acid; BR, brassinosteroid; ETH, ethylene. The circle, triangle, and square indicate cellular metabolism under flooding, drought, and salt, respectively. The orange color indicates activated metabolism under indicated environmental stressors. The solid and dashed lines indicate the confirmed and elusive interactions within phytohormones, respectively, in stressed soybeans.

**Figure 5 plants-12-02865-f005:**
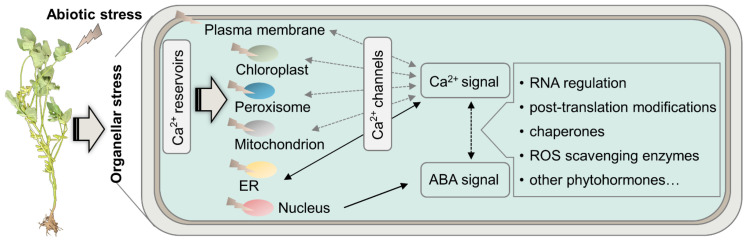
Proposed stress signals of calcium and ABA in the generic response to abiotic stress based on organellar stress in soybeans. Abiotic stress is sensed by soybean, followed by calcium signals generated from calcium reservoirs in plant cells. Nuclei stress in response to environmental stimuli generates an ABA signal and could integrate with calcium signals to trigger the changes in RNA regulation, protein translation modifications of proteins, phytohormone signals, and expression of chaperones involved in protein folding and enzymes for ROS scavenging.

**Table 1 plants-12-02865-t001:** Subcellular proteomics employed in soybeans under abiotic stress.

Subcellular	Organ	Plant	Stress	Technique	Findings	Ref.
ER	root tip	2-day-old	flood/2 days	1DE LC–MS/ MS, gel-free LC–MS/MS	Under flood conditions, 117 and 212 proteins increased and decreased, and 111 proteins were enriched in post-translational modification, protein synthesis, folding, degradation, and activation.	[114]
ER	root tip	2-day-old	flood/2 days, drought/2 days	gel-free/label-free LC–MS/MS	A number of 368 and 103 proteins were identified under flood conditions and drought, and proteins enriched in protein glycosylation and signal responded to both stresses.	[27]
Chloroplast	leaf	10-day-old	O_3_ (120 ppb)/3 days	2DE MALDI–TOF/MS	A total of 32 proteins were responsive to ozone stress, and proteins associated with photosynthesis mostly decreased.	[23]
Mitochondrion	root and hypocotyl	2-day-old	flood/2 days	2DE, BN-PAGE LC–MS/MS	In mitochondria, 34 matrix proteins and 16 membrane proteins changed by flood conditions, and proteins related to the tricarboxylic acid cycle increased, while inner membrane carriers and proteins related to complexes III, IV, and V of electron transport chains decreased.	[115]
Mitochondrion	root tip	2-day-old	flood with A1_2_O_3_ (5, 30–60, 135 nm at 50 ppm)/2 days	gel-free/ label-free LC–MS/MS	Under flood conditions, six, five, and six proteins were specifically changed in plants exposed to 5, 30–60, and 135 nm A1_2_O_3_, indicating various sizes of nanoparticles affected membrane permeability and the tricarboxylic acid cycle activity through mediating mitochondrial proteins.	[25]
Nucleus	root tip	2-day-old	flood/3, 6, 24 h	gel-free/label-free LC–MS/MS	A total of 237 proteins were altered by flood conditions in a time–course manner, while 365 proteins were changed by the initial 3 h flood and mapped to pre-mRNA processing and pre-ribosome biogenesis.	[116]
Nucleus	root tip	2-day-old	flood/3 h	gel-free/label-free LC–MS/MS	Under flood conditions, 14 nuclear phosphoproteins were changed, and phosphorylation of zinc finger/BTB domain-containing protein 47 was responsive to activated ABA signal for stress tolerance.	[117]
Plasma membrane	leaf and root	7-day-old	NaCl (50, 100 mM)/7 days	gel-free/label-free LC–MS/MS	Under the presence of salt, 140 and 57 proteins were specific to the root and leaf, and they were mainly involved in transport inside the cell.	[118]
Plasma membrane	root tip	14-day-old	AlCl_3_ (50 μM in 0.5 mM CaCl_2_)/3 days	TMT LC–MS/MS	A total of 90 proteins were differentially accumulated and enriched in membrane trafficking/transporters, cell wall modification, defense response, and signal transduction.	[119]
Cell wall	root and hypocotyl	2-day-old	flood/2 days	2DE MALDI–TOF/MS	Under flood conditions, 16 proteins responded with decreased stem 28/31 kDa glycoprotein precursors, germin-like protein precursors, and increased methionine synthases.	[120]
Peroxisome	cotyledon	seed	dark/7 day	2DE MALDI–TOF/MS	Identified peroxisomal proteins harboring peroxisomal targeting signal sequences were associated with the beta-oxidation cycle, glyoxylate cycle, and stress response.	[121]
Peroxisome	cotyledon	seed	dark/7 day	BN-PAGE MALDI–TOF/MS	GmPNC1 was identified, and PNC1 contributed to the transport of adenine nucleotides consumed during post-germinative growth.	[122]

## Data Availability

Not applicable.

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
