# Peer review of "Subcellular Proteomics to Elucidate Soybean Response to Abiotic Stress"

_plants, 2023, doi:10.3390/plants12152865_

Round 1

Reviewer 1 Report

Wang and Komatsu's review is adequate and timely. The text explores various types of abiotic stress, although it is biased towards flood stress. In addition, there are several self-citations, but they are appropriate given the authors' contribution to the field. A key point that the authors do not address is the very concept of spatial proteomics. In fact, what the review does address is the concept of organellar proteomics which is not the same as spatial proteomics, if the spatial concept is seen as the proteomics of a tissue or cells in a given position. Based on this, the title of the review could also be revised. The review can be a contribution to researchers working on abiotic stress in soybean.

As minor points, please revise:

Please check text
126 Progressive achievements have been obtained in
127 soybean tolerance to salt, while studies on saline-drought and -alkali need emphasize, .
128 which soybean always experience for field cultivation.

Please check text
158 During vegetative stage, even 3-h heat could induce alternations of protein levels in root
159 hairs and stripped roots, and cell wall formation, amino acid metabolism, lipid
160 biosynthesis, and ROS scavenging were enriched by heat-induced proteins in soybean
161 within 24-h exposure [101].

Please revise the mention of calnexin on Figure 3 which is duplicated.

Please describe which nanoparticles
382 Both of the abundance and enzyme activities of glutathione-S-transferase
383 and lipoxygenase in mitochondria was increased by flooding stress, along with the
384 increase of hydrogen peroxidase [146], and reduced ascorbate/glutathione cycle was
385 related to reduced ROS scavenging activity upon nanoparticle application [25].

There are a number of sentences that need to be revised to make the text more readable for readers. Otherwise, it will be difficult to grasp the ideas explored by the authors.

Author Response

Reviewer 1

Wang and Komatsu's review is adequate and timely. The text explores various types of abiotic stress, although it is biased towards flood stress. In addition, there are several self-citations, but they are appropriate given the authors' contribution to the field. A key point that the authors do not address is the very concept of spatial proteomics. In fact, what the review does address is the concept of organellar proteomics which is not the same as spatial proteomics, if the spatial concept is seen as the proteomics of a tissue or cells in a given position. Based on this, the title of the review could also be revised. The review can be a contribution to researchers working on abiotic stress in soybean.

Answer: Thank you very much for your suggestion. Based on comments from two reviewers, the title has been changed into “Subcellular Proteomics to Elucidate Soybean Response to Abiotic Stress”

As minor points, please revise:

Please check text

126 Progressive achievements have been obtained in soybean tolerance to salt, while studies on saline-drought and -alkali need emphasize, which soybean always experience for field cultivation.

Answer: Thanks for the comment. This sentence has been corrected as follows: “Progressive achievements of soybean response to salt stress have been obtained, whereas the episode of drought or alkali always happens in an occurrence with salt in the field, therefore, studies on plant response to binary stimuli need emphasize.”

Please check text

158 During vegetative stage, even 3-h heat could induce alternations of protein levels in root hairs and stripped roots, and cell wall formation, amino acid metabolism, lipid biosynthesis, and ROS scavenging were enriched by heat-induced proteins in soybean within 24-h exposure [101].

Answer: This sentence has been changed into “For 3-day-old seedlings, even 3-h heat was sufficient to trigger significant changes at protein level in root hairs and stripped roots, and stress-related proteins played roles in chromatin remodeling and post-transcription regulation in root hairs within 24-h exposure to heat [101].”

Please revise the mention of calnexin on Figure 3 which is duplicated.

Answer: We are sorry for this mistake. The duplicated calnexin has been removed.

Please describe which nanoparticles

382 Both of the abundance and enzyme activities of glutathione-S-transferase and lipoxygenase in mitochondria was increased by flooding stress, along with the increase of hydrogen peroxidase [146], and reduced ascorbate/glutathione cycle was related to reduced ROS scavenging activity upon nanoparticle application [25].

Answer: Thank you very much for the comment. The information of aluminum oxide nanoparticles has been added in red.

There are a number of sentences that need to be revised to make the text more readable for readers. Otherwise, it will be difficult to grasp the ideas explored by the authors.

Answer: As required, manuscript writing has been improved for revision in red.

Reviewer 2 Report

Suggestions,

1)    The review is about spatial proteomics, but I don’t see any information on proteomics in abstract.  Abstract section needs extensive revision and re-writing. 

2)    The paper is poorly written.  The review doesn’t look attractive, authors need to highlight proteomics related studies (in tabular form) more as review is on proteomics. 

3)    Extensive revision for English language is needed.

4)    Why soybean crop?

5)    Why proteomics approach? Why not transcriptomics? Authors has highlighted few gene function related studies. Don’t you think you should change title to molecular mechanism?

6)    Line 488, how bioinformatics will progress the understanding? Please elaborate more on this.

7)    Discuss future prospectus.

8)    Discuss losses due to abiotic stresses in soybean? Discuss with specific data?

9)    Authors need to add some interesting figures to add value to review.

Extensive revision for English language is needed.

Author Response

Reviewer 2

  1. The review is about spatial proteomics, but I don’t see any information on proteomics in abstract. Abstract section needs extensive revision and re-writing. 

Answer: Thank you very much for your suggestion. Based on the comments from two reviewers, the manuscript title has been changed into “Subcellular Proteomics to Elucidate Soybean Response to Abiotic Stress”. In addition, the abstract has been rewritten in red.

  1. The paper is poorly written. The review doesn’t look attractive, authors need to highlight proteomics related studies (in tabular form) more as review is on proteomics. 

Answer: The manuscript writing has been improved. Subcellular proteomic-related studies have been prepared in Table 1 in revision.

  1. Extensive revision for English language is needed.

Answer: English language of this manuscript has been improved for revision in red.

  1. Why soybean crop?

Answer: Thank you very much for the comment. The abundant nutrients, human health promotion, and disease prevention make soybean specific compared with other crops. These important values of soybean have been described in Introduction in red. In addition, numerous studies of subcellular proteomics have been conducted in soybean compared with other crops and it is advantageous to reveal plant response to abiotic stress on organellar level.

  1. Why proteomics approach? Why not transcriptomics? Authors has highlighted few gene function related studies. Don’t you think you should change title to molecular mechanism?

Answer: Thank you very much for these questions. The reason using subcellular proteomics has been added in Introduction in red. Since molecular mechanism may exclude the physiological response of organellar stress enriched by stress responsive proteins in soybean, the title has not been changed to molecular mechanism, and finally it has been corrected as “Subcellular Proteomics to Elucidate Soybean Response to Abiotic Stress” based on the comments from all reviewers.

  1. Line 488, how bioinformatics will progress the understanding? Please elaborate more on this.

Answer: Thanks for the comment. The elaboration of bioinformatics in progressing our understanding on soybean stress response has been added in red.

  1. Discuss future prospectus.

Answer: As commented, discussion of future prospects has been added in red.

  1. Discuss losses due to abiotic stresses in soybean? Discuss with specific data?

Answer: Thanks for the comment. The yield loss of soybean due to abiotic stress has been informed in Concluding remarks in red.

  1. Authors need to add some interesting figures to add value to review.

Answer: Thank you very much for the comment. New figure 5 has been prepared to inform the generic reaction of organellar stress of soybean under various environmental conditions based on the results of subcellular proteomics.

  1. Extensive revision for English language is needed.

Answer: English language has been polished in revision in red.

Round 2

Reviewer 1 Report

Authors address all my previous concerns.

English must be revised, please.

Author Response

Reviewer 1

Authors address all my previous concerns.

Answer: Based on your suggestion, this article could be improved. Thank you very much for your suggestions and cooperation.

English must be revised, please.

Answer: English language has been edited by native English speaker in blue.

Reviewer 2 Report

I don't think the revision improved the publication standard. It still appears to be a rough draft. Major edition of English language is needed. Please take help of native English speaker. The review is not going with flow of story.  Many changes still have to be made. Please re-consider.

Major edition of English language is needed. Please take help of native English speaker.

Author Response

Reviewer 2

I don't think the revision improved the publication standard. It still appears to be a rough draft. Major edition of English language is needed. Please take help of native English speaker.

Answer: Thank you very much for the comments. English language has been edited by native English speaker in blue.

The review is not going with flow of story. Many changes still have to be made. Please re-consider.

Answer: Thank you very much for the comments. The workflow of this manuscript has been described in Introduction in blue.

Round 3

Reviewer 2 Report

The review has improved as compared to previous version. I endorse it for publication.

Minor editing of English language required

Author Response

Reviewer 2

Minor editing of English language required.

Answer: Thank you very much for your additional comments. This article has been modified by a native English speaker, again.